# The Replacement of Fish Meal with Poultry By-Product Meal and Insect Exuviae: Effects on Growth Performance, Gut Health and Microbiota of the European Seabass, *Dicentrarchus labrax*

**DOI:** 10.3390/microorganisms12040744

**Published:** 2024-04-06

**Authors:** Simona Rimoldi, Ambra Rita Di Rosa, Rosangela Armone, Biagina Chiofalo, Imam Hasan, Marco Saroglia, Violeta Kalemi, Genciana Terova

**Affiliations:** 1Department of Biotechnology and Life Sciences, University of Insubria, 21100 Varese, Italy; simona.rimoldi@uninsubria.it (S.R.); ihasan@uninsubria.it (I.H.); marco.saroglia@gmail.com (M.S.); vkalemi@uninsubria.it (V.K.); 2Department of Veterinary Sciences, University of Messina, 98168 Messina, Italy; ambra.dirosa@unime.it (A.R.D.R.); rosangela.armone@studenti.unime.it (R.A.); biagina.chiofalo@unime.it (B.C.)

**Keywords:** insect meal, exuviae, poultry meal, seabass, gut microbiota

## Abstract

This study addressed the urgent need for sustainable protein sources in aquaculture due to the depletion of marine resources and rising costs. Animal protein sources, particularly poultry by-product meal (PBM) and insect exuviae meal, were investigated as viable alternatives to fishmeal (FM). The research study confirmed the successful replacement of FM with a combination of PBM and insect exuviae meal (up to 50%) in the diet of European seabass without compromising growth, feed conversion, gut health, and liver fat content. In particular, growth was robust with both PBM formulations, with the 25% PBM diet showing better results. Histological examinations showed good gut and liver health, contradicting the concerns of previous studies. This paper emphasizes the importance of holistic analyzes that go beyond growth parameters and include histomorphological investigations. The results show that PBM in combination with insect/exuviae meal is well tolerated by seabass, which is consistent with reports in the literature of it mitigating negative effects on gut health. A detailed analysis of the microbiota revealed a decrease in the Firmicutes/Proteobacteria ratio due to an increase in potentially pathogenic bacteria. However, the formulation containing insect exuviae partially counteracted this effect by preserving the beneficial Lactobacillus and promoting the synthesis of short-chain fatty acids (SCFAs), particularly butyrate. Chitin-rich components from insect exuviae were associated with improved gut health, which was supported by the increased production of SCFAs, which are known for their anti-inflammatory properties. This paper concludes that a combination of PBM and insect/exuviae meal can replace up to 50% of FM in the diet of seabass, supporting sustainable aquaculture practices. Despite some changes in the microbiota, the negative effects are mitigated by the addition of insect exuviae, highlighting their potential as a prebiotic to increase fish productivity and contribute to a circular economy in aquaculture.

## 1. Introduction

Fishmeal (FM) is undoubtedly an excellent source of protein in aquafeeds due to its essential amino acid (EAA) profile, palatability, and other remarkable properties [1]. However, in recent years, we have observed a steady decline in the inclusion levels of FM in aquaculture feeds due to limited supply, increasing price, and ethical concerns [1,2]. Currently, FM is widely regarded as a no longer sustainable and essential component of fish feed.

Therefore, alternative, and more sustainable protein sources are needed for the development and promotion of sustainable aquaculture. Since the beginning of this century, considerable research efforts have been made to find alternative proteins, focusing mainly on plant-based ingredients such as soybean meal, corn gluten meal, and rapeseed meal. Plant proteins are used extensively in feed formulations and will continue to be important raw materials in aquafeeds, although they contain several antinutritional factors (ANFs), complex indigestible carbohydrates, and low EAAs, which can have negative side effects on feed intake, the digestion and absorption of nutrients, and fish health [3].

A real alternative to vegetable proteins as FM substitutes are animal by-products, which have similar properties to FM in terms of AA content, digestibility, palatability and lack of ANFs. Among the by-products of land animals, poultry by-product meal (PBM), which consists of ground, rendered, and cleaned parts of the carcasses of slaughtered poultry, including legs, necks, intestines, and undeveloped eggs, is the most economical and widely used component of aquafeed [4]. PBM is a high-quality palatable and digestible protein source due to its content of EAA (except lysine and methionine), fatty acids, vitamins, and minerals [5].

To date, the suitability of PBM inclusion has been reported for several marine species [6]. PBM has successfully partially or completely replaced FM in the diet of several marine fish species, including gilthead seabream (*Sparus aurata*) [7,8,9,10], black seabass (*Centropristis striata*) [11], and red seabream (*Pagrus major*) [12]. In gilthead seabream, up to 100% of FM replacement (corresponding to 38% of inclusion) was achieved with lysine and methionine supplementation [10,13].

Optimal growth performance has even been obtained in salmonids, including rainbow trout (*Oncorhynchus mykiss*) and Atlantic salmon (*Salmo salar*) fed feeds with high PBM content [14,15,16,17,18]. However, it should be noted that differences in the maximum dietary PBM inclusion rate are highly dependent on fish species, PBM quality, and overall feed formulation. Although some studies suggest that PBM can completely replace conventional protein sources, most studies recommend partial replacement of FM to maintain the nutritional balance, palatability, and digestibility of the feed [6,7,8,9,15,19,20].

Moreover, mixtures of alternative protein sources are usually preferred over a single protein source to replace FM in fish feed, especially in FM-free formulations. For example, mixtures of poultry and insect meal are being investigated as alternatives to FM in aquafeeds. Like PBM, insect meal is a promising alternative to FM in aquafeed as it has several advantages. Insects have a high protein (34–74% DM) and lipid (10–30% DM) content, balanced EAA, and good content of vitamins (B12) and minerals (iron and zinc); they also contain several bioactive compounds such as chitin, fatty acids, and antimicrobial peptides [21].

The use of insects in animal feed is strictly regulated by the European Union (EU), which authorizes the use of eight insect species in feed for aquaculture, namely the common housefly (*Musca domestica*), the black soldier fly (*Hermetia illucens*), lesser mealworm (*Alphitobius diaperinus*), mealworm (*Tenebrio molitor*), house cricket (*Acheta domesticus*), field cricket (*Gryllus assimilis*), banded cricket (*Gryllodes sigillatus*), and silkworm (*Bombyx mori*). Among them, the black soldier fly and mealworm are the most used insect species in aquafeed [22,23].

Recent studies have shown that an effective replacement of FM in aquafeed can be achieved by combining black soldier fly (BSF) larval meal with PBM. Indeed, BSF and PBM successfully replaced plant proteins in a diet without FM in gilthead seabream, improving nutrient uptake and promoting gut health [24].

In addition, the combination of BSF and PBM restored the gut microbiota of fish negatively affected by a plant-based diet by improving the richness of bacterial species and abundance of beneficial bacteria [25,26]. This prebiotic effect is principally due to chitin, the indigestible polysaccharide that makes up the exoskeleton of insects [22,27]. The addition of insect meal usually increases the amount of lactic acid bacteria (LAB) and the genus Bacillus, which are commonly used as probiotics in aquaculture. Finally, the gut microbiota of fish is known to be very sensitive to dietary manipulations [28], and since proteins are the most important nutrient, they have a major impact on the composition of the gut microbiota.

Accordingly, in the present study, we tested the effect of the partial replacement of FM by PBM alone or in combination with insect meal or exuviae meal, a chitin source from insect exoskeletons, in European seabass. A multidisciplinary approach was used in which growth performance, gut and liver morphology, gut microbiota, and gut volatile short chain fatty acid production were evaluated.

## 2. Materials and Methods

### 2.1. Ethics Approval

The handling of animals and all procedures complied with the European Union Council Directive 2010/63 on experimental animals. The protocol was approved by the Ethics Committee for Animal Welfare and Use of Animals of the University of Insubria and by the Italian Ministry of Health (No. 285/2020-PR).

### 2.2. Diets

Three isoproteic and isolipidic diets were formulated: a control diet (A) containing 20% FM and two test diets in which 50% of the FM was replaced by 20% (B) or 25% (C) PBM. In addition, 5% insect meal (Mutatec, Cavaillon, France) and 0.5% exuviae meal were added to feed B and C, respectively. The complete formulation and proximate composition of the three feed types produced by “Leocata Mangimi” in Modica (Ragusa), Italy, are listed in Table 1 and Table 2.

### 2.3. Feeding Trial and Sampling

The feeding experiment was carried out in a recirculating aquaculture system (RAS) at the Department of Biotechnology and Life Sciences of the University of Insubria (Varese, Italy). Two hundred and ten juvenile European seabass (*D. labrax*) (mean initial body weight 52 ± 0.8 g), purchased from Società Agricola CIVITA ITTICA S.r.l. (Verona, Italy), were randomly distributed in six circular fiberglass tanks with a capacity of 700 L (35 fish/tank). The water’s pH, temperature, and dissolved oxygen (DO) were strictly controlled throughout the experiment. Water temperature was 20 ± 1.5 °C, salinity was 22 g L^−1^, and pH was 7.5–8.0; total ammonia nitrogen was ≤0.1 mg/L, and DO saturation was maintained above 85%. After a one-week acclimatization period during which the fish were fed a commercial diet (NEO STEP2, VRM srl—Naturalleva, Verona, Italy), the fish were fed three experimental diets (2 tanks/feed) once a day (6 days a week) for 15 weeks (about 3 and a half months). The experimental groups fed diets A, B, or C were named CTRL, PM20, and PM25, respectively. The fish were individually weighed and measured by length at the beginning of the feeding trial (t0) and then 4 weeks (t1) and 10 weeks after (t2) and at the end of 15 weeks (t3). The feeding rate was adjusted according to biomass and ranged between 1.5 and 2.5% of body weight. The weight of ingested and uneaten feed was recorded daily for each tank. Mortality was also monitored. These data were used as the basis for calculating the feed conversion ratio (FCR = dry feed intake/wet weight gain) and specific growth rate [SGR (%/day) = 100 × [ln (final body weight) − ln (initial body weight)]/day] for each dietary fish group.

At the end of the 16-week experiment, 4 fish per tank (8 fish/diet) were euthanized with an overdose of tricaine methanesulfonate (MS-222, 400 mg/L, Sigma-Aldrich, St. Louis, MO, USA). Liver, proximal, and distal intestine were removed from each fish and immediately fixed in a 10% solution of neutral buffered formalin (NBF) for histological analysis. Samples were taken from an additional four fish per tank (8 fish/diet), and the entire intestine (excluding pyloric ceca) was aseptically removed from each fish. The digesta and intestinal mucosa were collected and mixed in a sterile tube with 800 μL of Xpedition™ lysis/stabilization solution (Zymo Research, Irvine, CA, USA) and stored at 4 °C until metabarcoding analysis. For the quantification of volatile fatty acids, 18 additional fecal samples were collected from each experimental group (9 fish/tank), pooled in a centrifuge tube (3 pools/tank), and stored at −80 °C until analysis.

### 2.4. Histological Analysis

NBF-fixed liver and intestinal samples were embedded in paraffin and cut into 5 μm sections using a microtome (Leica RM2245, Leica Biosystems, Milan, Italy). To observe the tissue structure, the slides were stained with hematoxylin and eosin (H&E) and examined under a light microscope (Zeiss Axiophot microscope, Milan, Italy) using a CMOS Discovery C30 digital camera. The acquired images were processed using Fiji software (open-source Java-based image processing program, https://fiji.sc/, accessed on 15 January 2024). For intestinal morphology, villus height (ViH), villus width (ViW), lamina propria width (LPW), and submucosal layer thickness (SMT) were measured according to Escaffre et al. [29]. For the liver, a semi-quantitative assessment approach based on a grading score (1 = not observed/low, 2 = moderate, and 3 = severe) was used [30,31,32]. Hepatocyte vacuolization (HV), nuclear displacement (ND), cellular hypertrophy (CH), and irregular nuclear shapes (NSs) were considered as histological features in the liver.

### 2.5. Analysis of Short-Chain Fatty Acids in Faecal Samples

The qualitative and quantitative determination of SCFAs (acetate, propionate, iso-butyrate, and butyrate) was performed according to the modified extraction method of Chlebicz-Wójcik and Śliżewska [33]. The detailed protocol of SCFA extraction was described by Rimoldi et al. [34]. The extracts were analyzed using an HPLC-UV-VIS system (Shimadzu, Milan, Italy) equipped with two LC-20AD pumps, a CBM-Alite controller, a DGU-20A5 degasser, and an automatic injector. Samples were separated on a 150 mm × 4.6 mm I.D., 2.7 µm particle Ascentis Express 90A C18 column (Merck KGaA, Darmstadt, Germany). The flow rate of the mobile phase (0.005 M H_2_SO_4_) was 0.6 mL/min, and the injection volume was 10 µL. The column temperature was maintained at 60 °C, and the UV-VIS absorbance was measured at a wavelength of 210 nm. The raw data were processed using LC software (LabSolutions Single LC S/N L52405100502LG, Shimadzu, Milan, Italy). The peaks were identified based on the retention time of the certified standards. Quantitative analyses were performed using the calibration curve method in a range of different concentrations. The results were expressed in mmol/L.

### 2.6. Bacterial DNA Extraction, Multiplex Amplicon Library Preparation and Sequencing

Bacterial DNA was extracted from 300 mg of intestinal material (faeces + mucosa) and 200 mg of each feed in triplicate. For DNA extraction, the DNeasy^®^ PowerSoil^®^ Pro kit (Qiagen, Milan, Italy) was used according to the manufacturer’s instructions, with an additional mechanical lysis step using a TissueLyser II (Qiagen, Italy). The 16S V4 library was prepared by the next-generation sequencing (NGS) service GALSEQ srl (Milan, Italy) and sequenced on the NovaSeq 6000 System—Illumina (San Diego, CA, USA)—using a paired-end 2 × 150 bp sequencing strategy and cluster density of 300 K/sample. A detailed description of library preparation and sequencing has already been reported [16,35].

### 2.7. NGS Raw Data Analysis

The QIIME 2^TM^ pipeline (v. 2020.2) was used to process raw amplicon data [36]. Taxonomy was assigned to amplicon sequence variants (ASVs) down to the genus level using the SILVA database (https://www.arb-silva.de/, accessed on 7 January 2024). The entire data preparation workflow has been previously described in detail [35]. It included pre-processing steps, taxonomy classification, and calculation of alpha and beta diversity based on weighted and unweighted UniFrac distances [37,38].

### 2.8. Predicted Functional Pathway Analysis of Gut Microbiome

PICRUSt (Phylogenetic investigation of communities by reconstruction of unobserved states) was used to assess the functional potential of microbial communities [39]. The predicted function of the metagenome was based on the analysis of KEGG pathways. The extended error bar graphs showing the significantly different abundance of KEGG pathways between control and PM20 and PM25 groups were generated from the PICRUSt output files using the Statistical Analysis of Metagenomic Profiles (STAMP) software package (Ver 2.1.3, 26 June 2015, accessed on 7 January 2024) and two-sided Welch’s *t*-test [40].

### 2.9. Statistics

All data were tested for normality and homoscedasticity using the Shapiro–Wilk test and Levene’s test, respectively. For statistical comparisons, the two-way analysis of variance or Kruskal–Wallis test were used if the data were not normally distributed. Significant differences in beta diversity were tested using permutational multivariate analysis of variance (PERMANOVA). All tests were performed using Past4 v. 4.02 software with the significance set at *p* < 0.05 [41]. Differences in the abundance of bacterial taxa between samples were tested with the two-sided Welch’s *t*-test using the STAMP software package.

## 3. Results

### 3.1. Growth Performance

The data on growth performance and feed conversion are shown in Table 3. During the entire duration of the feeding trial (88 days), the mortality rate was less than 2%. All fish grew efficiently and doubled their weight. However, a significantly lower final body weight (FBW) was observed in fish fed diet PM20 with respect to the controls. There were no significant differences between the diet groups in specific growth rate (SGR) and feed conversion ratio (FCR).

### 3.2. Gut and Liver Morphology

A standard histological analysis of proximal and distal intestinal cross-sections of the PM20 and PM25 fish groups showed a well-organized and preserved tissue structure with no obvious signs of damage or inflammation compared with specimens from control animals (Figure 1A–F). A morphometric analysis of the intestinal cross-sections also showed no morphological changes in response to diet in the structure of the mucosa and submucosa layers, thickening of the lamina propria, and height and width of the villi in both the distal and proximal intestines (Table 4).

Histological sections of the liver are shown in Figure 2A–F. The liver tissue samples yielded a total score of six for the control group, five for the PM20 group, and 5.67 for the PM25 group (Table 5). These values define a normal or slightly altered morphology. Accordingly, all samples showed moderate lipid infiltration, which did not affect the size and shape of the hepatocytes. Most of the nuclei were in a peripheral position but did not alter the shape and size of the hepatocytes and thus the physiological morphology of the organ. No signs of hepatocyte ballooning, vacuolar degeneration, capillary hyperemia, or vasodilatation were observed in the different experimental groups.

### 3.3. Volatile SCFAs in Faecal Samples

The SCFAs in the feces of fish fed three experimental diets were quantified by HPLC. The concentrations of selected SCFAs (acetate, propionate, and butyrate) found in our samples are shown in Table 6. The highest amount of propionate (7.41 mmol/L) and butyrate (3.03 mmol/L) was found in the gut of fish of group PM25 fed with insect exuviae. In contrast, fish of the PM20 group fed with diet B had a higher concentration of acetate in their feces compared with the other two feeding groups.

### 3.4. Sequencing Efficiency

Of the bacterial DNA extracted from the gut samples, only twenty were efficiently amplified to obtain a V4 amplicon library, with seven each from the CTRL and PM25 fish groups and six from the PM20 group. In contrast, all nine DNA samples from the feeds were amplified correctly and yielded the expected amplicon size. All twenty-nine samples were successfully sequenced and yielded 2,393,031 high-quality reads, of which 2,021,977 were from gut samples and 371,054 from feed pellets (Appendix A). Good’s coverage values were above 99% for all samples, indicating that the sequencing depth of all datasets in this study was sufficient to reveal the bacterial communities in the gut and feed. Based on the rarefaction curves, the sequencing depth for the calculation of alpha diversity indices was set at 55,556 reads.

All raw data (Fastq) were submitted to the public database of the European Nucleotide Archive (EBI ENA) under the access code PRJEB70800.

### 3.5. Feed Microbial Profiles

The microbiota profile of the three feeds was outlined at the level of phylum, class, order, family, and genus. Considering only the most representative taxa (relative abundance ≥ 1%), the microbial community of the feeds was composed of 4 phyla (Figure 3A), 5 classes, 17 orders, 59 families (Figure 3B), and 73 genera. The result of the alpha diversity analysis performed on the feed microbiota data showed the highest species richness (observed OTUs) and biodiversity (Shannon’s index) in feed A and a lower Chao 1 index value in feed C compared with B (Table 7).

Pairwise comparisons of microbial communities from different diets using the two-sided Welch’s *t*-test revealed that 37 and 16 genera differed significantly between control A and diets B and C, respectively. Most of these genera belonged to the phylum Firmicutes and were more abundant in the experimental feed pellets (Appendix A. In contrast, the relative abundances of 52 bacterial genera were significantly different between the two experimental feeds B and C (Appendix A). Twenty-two of them were assigned to the Firmicutes phylum. In particular, the probiotic genera *Bacillus* and *Lactobacillus* were associated with diet C, which contained 25% PBM and exuviae meal in its formulation.

### 3.6. Gut Microbiota Profiles

The total intestinal microbiota of our samples comprised 5 phyla (Figure 4A), 5 classes, 13 orders, 46 families (Figure 4B), and 49 genera.

Analysis of the alpha diversity of the intestinal samples (Table 8) revealed significant differences only for the Shannon biodiversity index, which was lower than in fish fed diet B than in controls (CTRL = 6.20 ± 1.06; PM20 = 4.69 ± 0.65; PM25 = 5.18 ± 1.23).

Analysis of the beta diversity of gut microbial communities based on phylogenetic UniFrac distances revealed both qualitative (unweighted UniFrac) and quantitative (weighted UniFrac) differences as a function of diet. As shown in the PCoA diagrams in Figure 5A,B, the gut samples formed separate groups from the feed samples. In particular, the PCoA diagram based on the weighted UniFrac distances showed that the gut samples of fish fed the experimental diets PM20 and PM25 and the control samples were clustered separately (Figure 5A). The results of the PCoA analysis were statistically validated by both the ANOSIM test and PERMANOVA test (Table 9).

### 3.7. Modulating Effect of Diet on the Composition of Gut Microbiota

To determine any differences in the gut microbiota, pairwise comparisons were made between the feeding groups. The relative abundances of bacterial genera were compared using Welch’s *t*-test, and the results are shown in Table 10 and Table 11. Firmicutes and Proteobacteria are the dominant phyla in all experimental groups (Figure 4A). These two phyla were significantly affected by the replacement of FM with PBM. A decrease in Firmicutes (8–10%) combined with an increase in the proportion of Proteobacteria (80–83%) was observed in fish fed with experimental diets B and C compared with the controls (Firmicutes: 76%; Proteobacteria: 18%). At the genus level, a decrease in Lactobacillus was observed in the gut of PM20 (Table 11), but not in the PM25 group (Table 11). Compared with CTRL fish, both PM20 and PM25 fish had a lower relative abundance of the genera *Clostridium sensu stricto*, belonging to the Clostridiaceae family, and *Cetobacterium* (Table 10 and Table 11). In addition, the experimental diets B and C had a bactericidal effect on the genus Staphylococcus. In contrast, the PM20 and PM25 samples showed a marked increase in the genera Vibrio and Photobacterium, both of which belong to the Vibrionaceae family (Table 10 and Table 11). When comparing PM20 and PM25 fish, their gut microbiota differed only in the genus *Paenibacillus*, which was more abundant in the gut of PM20 fish (5%) than in the gut of PM25 fish (0.06%).

### 3.8. Predictive Functional Analysis of Gut Microbiota Communities

The PICRUSt predictions of the functional composition of the gut microbiota based on the KEGG database showed large differences in the abundance of KEGG metabolic pathways between the control and treatment groups. In particular, 20 and 19 metabolic pathways differed between the CTRL and PM20 and PM25 samples, respectively (Figure 6 and Figure 7). The gut microbiota of fish fed PBM diets showed increased abundance of metabolic pathways mainly involved in biofilm formation, peptidoglycan and lipopolysaccharide biosynthesis, structural proteins, the bacterial secretion system, and unsaturated fatty acids biosynthesis. In contrast, cell growth, nucleotide metabolism, and *Staphylococcus aureus* infections decreased in the PM20 and PM25 groups. No significant difference was found between the functional profile of PM20 and PM25.

## 4. Discussion

The drive to find alternative and more sustainable protein sources to replace FM stems from the need to alleviate the pressure on global marine resource depletion and to contain the cost of aquaculture production, as protein is the most expensive nutrient in fish diets, accounting for 30–50% of the total cost.

Various plant protein sources have traditionally been used to partially or completely replace FM in aquafeeds. However, nowadays animal protein sources are considered the best alternative to FM due to their higher protein and lipid content, better amino acid profile, and palatability [42,43].

The results of this study seem to proceed in this direction, confirming that PBM together with insect or exuviae meal successfully meets the requirements for adequate growth in seabass. This result confirms what has already been reported in the literature, which has suggested that the optimal substitution rate of FM by PBM is between 25% and 50% for most marine carnivorous animals [6]. Indeed, both formulations containing PBM were well accepted, and at the end of the feeding trial, the fish doubled their weight regardless of the diet.

However, only minor differences were found between the control group and the experimental feeding groups PM20 and PM25. The fish from the PM20 group performed poorly compared with those from the control group in terms of growth (final weight). On the other hand, no differences were found in the final weight of the fish from PM20 and PM25. In fact, in marine fish such as seabream (*Sparus aurata*) and red seabream (*Pagrus major*), there is evidence that even the complete replacement of FM with PBM had no negative effect on the growth parameters and productivity of the fish [8,12,13].

The present study is consistent with our previous findings in rainbow trout, a freshwater species [16]. In this previous study, trout fed a diet rich in PBM (55–70%) grew as well as fish fed a control diet rich in FM (37.3%) and free of PBM [16].

The formulation of optimal diets for farmed fish species requires the application of different types of analyses to verify their effects on the health status of the specimens, which cannot be limited to growth performance and feed efficiency alone. Histomorphological examination is a good biomarker for the assessment of fish welfare, especially in gut and liver histomorphology [44]. No potential changes related to intestinal inflammation or hepatic lipid accumulation in response to FM replacement were detected in the histological examinations, suggesting that PBM in combination with insect and exuviae meal was well tolerated by seabass. In agreement with our results, Pleić and colleagues [26] showed that partial replacement of plant protein diet with insect and PBM could even mitigate the negative effects of plant proteins on the proximal and distal gut of seabass by significantly improving all gut morphometric parameters, the degree of vacuolization, and cellular infiltration. The health and integrity of the intestinal epithelium are critical to nutrient absorption and fish health as damage to the gut can lead to immune dysfunction, reduced disease resistance, loss of appetite, and slow growth.

The liver is also considered a valuable marker for nutritional pathologies as it plays a key role in energy metabolism and storage as well as in immune defense and detoxification [45,46]. As for the present study, the histological results indicate a favorable liver health status, with a moderate accumulation of lipids in hepatocytes in all fish regardless of diet. In contrast, Donadelli et al. [47] recently observed a significant accumulation of hepatocyte lipids associated with marked histological changes that were indicative of an incipient steatotic state in seabass fed a FM-free diet in which 40% of the plant protein was replaced with insect or poultry by-product. These changes appeared to be slightly attenuated when insect meal and PBM were combined to partially replace plant proteins in the diet. The data available in the literature indicate species-specific differences in the responses to alternative protein-rich feed components. For example, the inclusion of *Hermetia illucens* and PBM in diets containing no FM resulted in improved gut and liver health in gilthead seabream and rainbow trout [18,24,47].

Insect meal and especially exuviae are considered as valuable sources of chitin. Chitin is an insoluble dietary fiber consisting of β-1,4-poly-N-acetyl-D-glucosamine and can be used as a substrate for bacterial fermentations, leading to the production of acetate, propionate, and butyrate as the main end products with positive effects on gut health [22,48]. In the present study, the highest amount of propionate and butyrate was found in the gut of seabass fed with exuviae meal. Accordingly, trout fed with pupal exuviae meal had the highest content of SCFAs, especially butyrate, in their feces [27]. Butyrate is well known to have anti-inflammatory properties and to promote fish intestinal health, barrier function, and mucosal immunity in fish. Therefore, its increased production in the fish gut should be considered a desirable effect [49,50].

As a prebiotic, chitin may also increase the biodiversity of the gut microbiota by promoting the proliferation of beneficial chitin-degrading bacteria, such as *Bacillus* and *Paenibacillus*, which have recently been isolated from the gastrointestinal tract of European seabass fed chitin-enriched diets [51]. The partial replacement of FM with at least 10% insect meal had an important effect in modulating the transient gut microbial communities by increasing both butyrate-producing bacteria and beneficial lactic acid bacteria [34,35,52,53,54]. Similarly, the ingestion of *H. illucens* exuviae meal led to an enrichment of the gut microbiota with the families Bacillaceae, Staphylococcaceae, Paenibacillaceae, and Brevibacteriaceae in seabass and rainbow trout [27,52]. Unlike previous studies, the proportion of insect or exuviae meal utilized in the present study was not sufficient to promote the proliferation of beneficial bacteria. Indeed, the gut microbiota of PM20 and PM25 seabass was characterized by an increase in the ratio between Proteobacteria and Firmicutes compared with the FM diet-fed controls. A similar increase in the Proteobacteria phylum, mainly represented by the Gammaproteobacteria class, was previously reported in trout fed diets containing a high proportion of alternative terrestrial animal proteins (>50%), mainly represented by PBM [16]. Accordingly, the trout gut microbiota was characterized by a high abundance of bacterial genera belonging to the class of Gammaproteobacteria, such as *Vibrio*, *Pasteurella*, and *Proteus*.

At the genus level, the PM20 and PM25 fish showed an enrichment of *Vibrio* and *Photobacterium* genera in their gut, which are normally considered potential pathogens for fish. Similarly, in a previous study, we found an increase in the genus *Photobacterium* in the gut microbiota of trout fed a diet containing 20% head shrimp meal, another chitin-rich ingredient [27]. However, it is also true that this genus includes several chitinase-producing bacterial species [55,56]. In contrast, it was recently reported that seabass fed a plant-based diet supplemented with a combination of 10% *H. illucens* meal and 30% PBM showed increased abundance of the phylum Firmicutes and, particularly, the beneficial genera *Lactobacillus* and *Bacillus* in their gut [26].

Here, *Lactobacillus* decreased significantly in the gut of PM20 seabass but not in PM25 compared with the controls. In contrast to the exuviae meal, the amount of insect meal was not sufficient to attenuate the negative effects of PBM on the abundance of lactic acid bacteria. Similarly, in gilthead seabream, a 10% content of *H. illucens* meal was not sufficient to increase the abundance of lactic acid bacteria in the gut [34,53].

The comparison of microbial profiles in the gut of PM20 and PM25 fish showed a positive association of the genus *Paenibacillus* with insect meal. This result confirmed what was previously reported in rainbow trout and gilthead seabream in response to dietary *H. illucens* meal ingestion [34,35,53]. In contrast, Rangel et al. [52] found that consumption of diets enriched with *H. illucens* exuviae, but not insect meal, led to an increase in the genus *Paenibacillus* in the gut of European seabass. This genus is of interest because it is considered a good probiotic candidate. In fact, *Paenibacillus* shares several characteristics with members of the genus *Bacillus.* It can produce antimicrobial and volatile organic compounds and degrade non-starch polysaccharides; last but not least, the bacteria of this genus have chitinolytic activity [51,52,57,58,59]. The presence of chitinolytic bacteria is particularly important when feeding insect-derived ingredients to fish as bacterial chitinases help to improve the digestibility of the feed [48]. In rainbow trout, on the other hand, the addition of 1.6% pupal exuviae meal of *H. illucens* to a diet containing 20% FM resulted in an enrichment of various bacteria genera, such as *Corynebacterium*, *Bacillus*, *Facklamia*, and *Brevibacterium*. These divergent results suggest that the relationship between dietary components and the gut microbiota are complex and species-specific.

The predictive functional analysis PICRUSt showed large differences between the controls and the PBM-fed experimental groups. Compared with the controls, fish fed PBM diets were associated with biofilm formation, peptidoglycan and lipopolysaccharide biosynthesis, structural proteins, bacterial secretion system, and unsaturated fatty acid biosynthesis. Much more interesting was the reduction of signaling pathways related to *Staphylococcus aureus* infection in PM20 and PM25 fish. This result is consistent with the lower amount of the genus *Staphyloccoccus* observed in these samples.

## 5. Conclusions

Overall, this work has shown that in practical feeds for European seabass formulated to current industry standards, FM can sometimes be replaced by up to 50% by a combination of PBM and insect or exuviae meal without compromising growth performance, feed conversion, gut health, and liver lipid content. The only negative effect of PBM was a decrease in the Firmicutes/Proteobacteria ratio at the gut microbiota level due to an increase in potentially pathogenic bacteria belonging to the Gammaproteobacteria class, such as Vibrio and Photobacterium. However, of the two experimental formulations, the diet containing insect exuviae meal partially mitigated this negative disadvantage by preserving the amount of the beneficial Lactobacillus genus comparable with controls fed FM diet and promoting the synthesis of SCFAs, especially butyrate. These data confirm our previous findings that insect exuviae are a valid prebiotic candidate to contribute to more sustainable aquaculture practices by increasing fish productivity through efficient utilization of organic waste, thus contributing to a circular economy system.

## Figures and Tables

**Figure 1 microorganisms-12-00744-f001:**
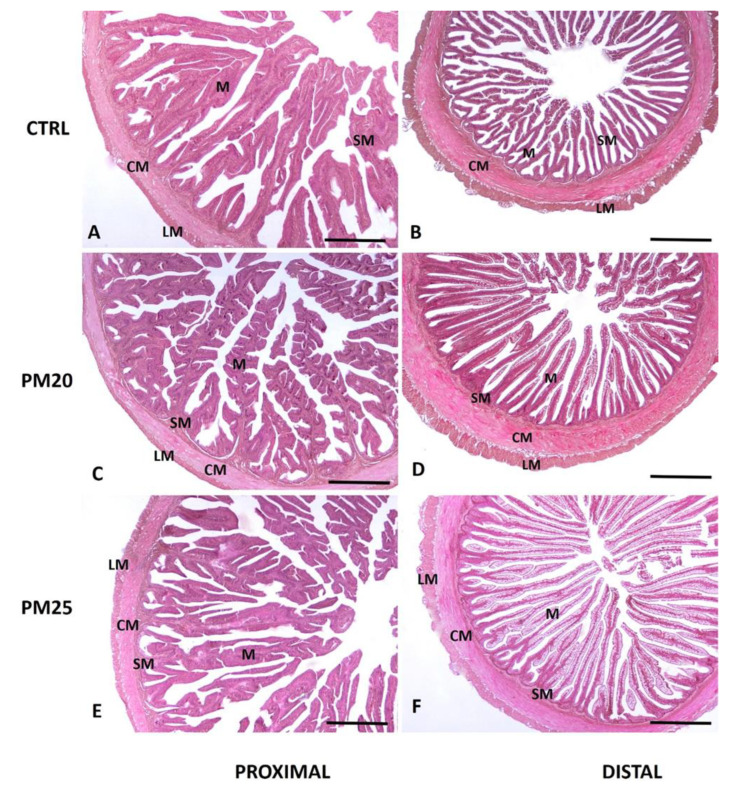
Standard hematoxylin-eosin (H&E) histochemical analysis of proximal (panels (**A**,**C**,**E**)) and distal (panels (**B**,**D**,**F**)) seabass intestine. We obtained 5 µm cross-sections from the proximal and distal paraffin-embedded intestine of CTRL (panels (**A**,**B**)), PM20 (panels (**C**,**D**)), and PM25 (panels (**E**,**F**)) fish. M, mucosa; SM, submucosa; CM, circular muscle layer; LM, longitudinal muscle layer. Scale bar = 1000 µm.

**Figure 2 microorganisms-12-00744-f002:**
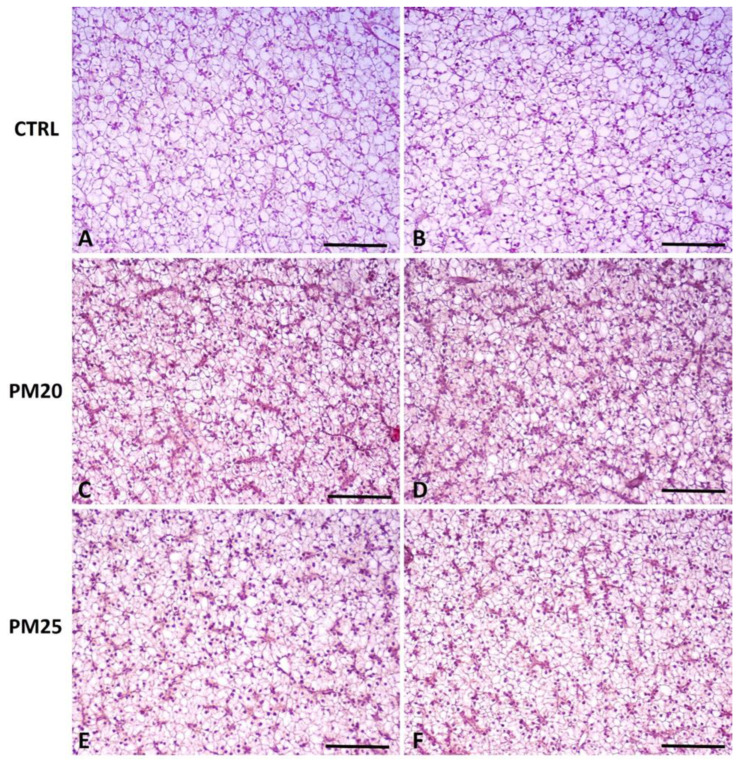
Standard haematoxylin–eosin (H&E) histochemical analysis of liver from representative images of fish fed control, PM20, and PM25 diets. We obtained 5 µm cross-sections from the paraffin-embedded liver of CTRL (panels (**A**,**B**)), PM20 (panels (**C**,**D**)), and PM25 (panels (**E**,**F**)) fish. Scale bar = 100 µm.

**Figure 3 microorganisms-12-00744-f003:**
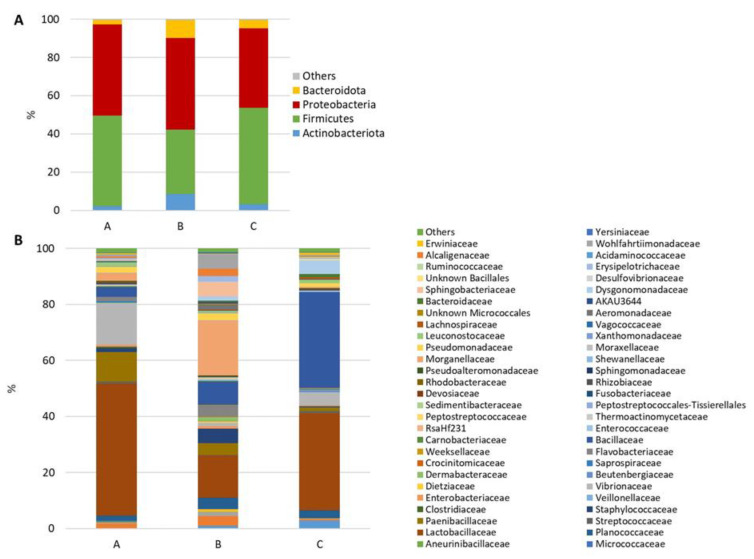
Stacked bar chart of the mean relative abundances (%) of the most abundant classified bacterial phyla (**A**) and families (**B**) most frequently found in feed samples.

**Figure 4 microorganisms-12-00744-f004:**
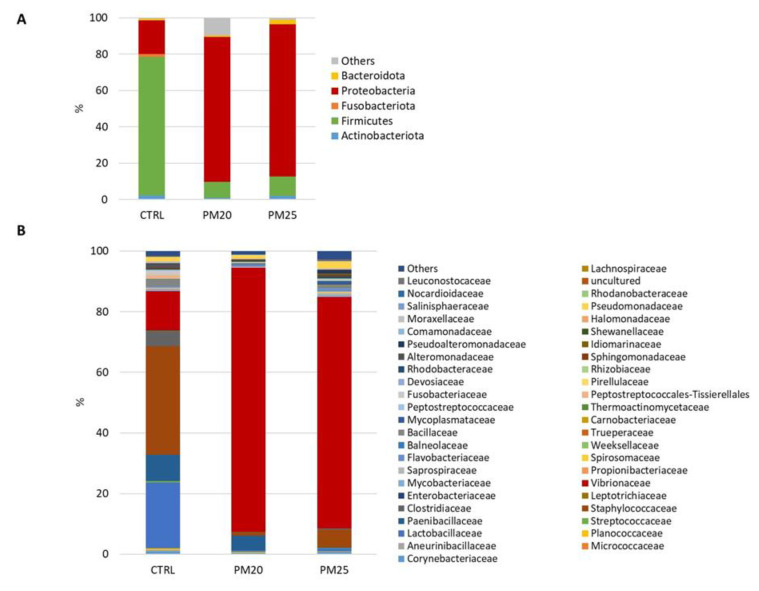
Stacked bar chart of the mean relative abundances (%) of the most abundant classified bacterial phyla (**A**) and families (**B**) found in gut samples.

**Figure 5 microorganisms-12-00744-f005:**
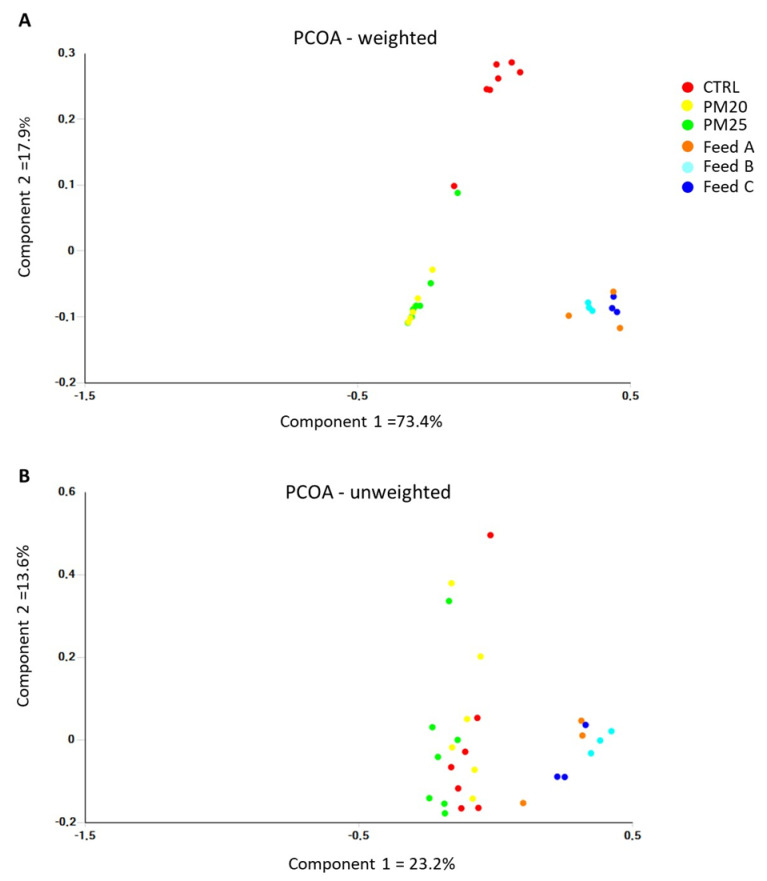
Plot of principal coordinate analysis (PCoA) using weighted (**A**) and unweighted (**B**) UniFrac distance matrices of gut microbial communities at the genus level.

**Figure 6 microorganisms-12-00744-f006:**
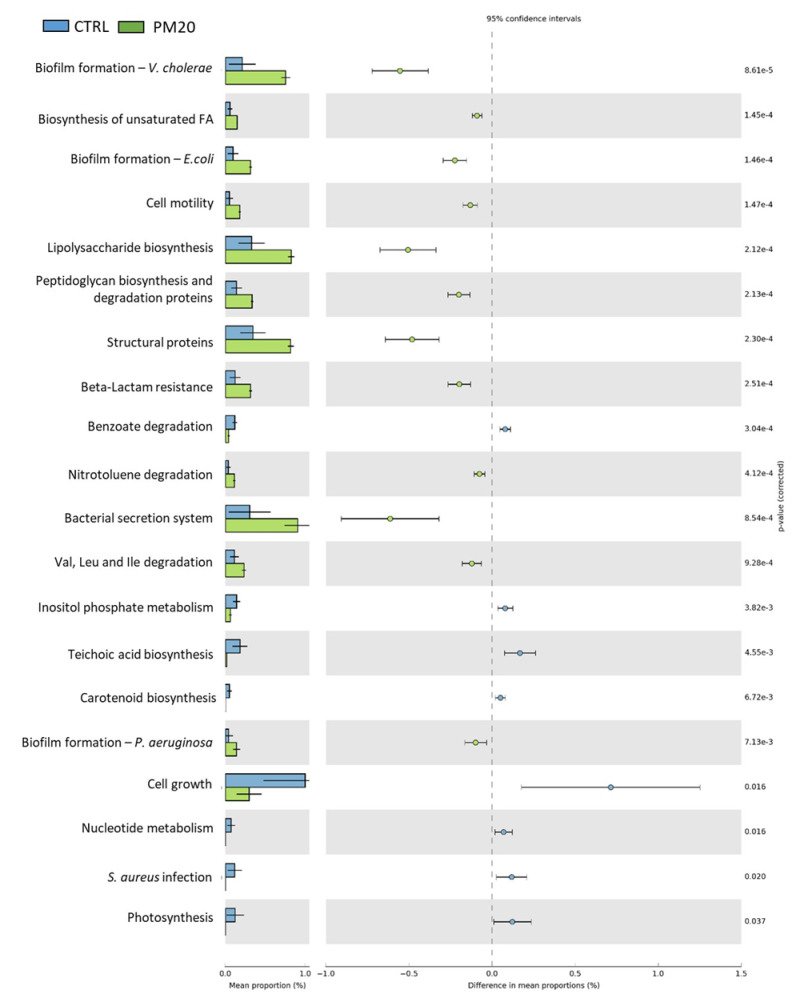
Comparison of the relative abundance of the PICRUSt functional profile of the gut microbiota between the control and PM20 experimental groups. Only the predicted functional pathways that differ significantly (*p* < 0.05) are shown.

**Figure 7 microorganisms-12-00744-f007:**
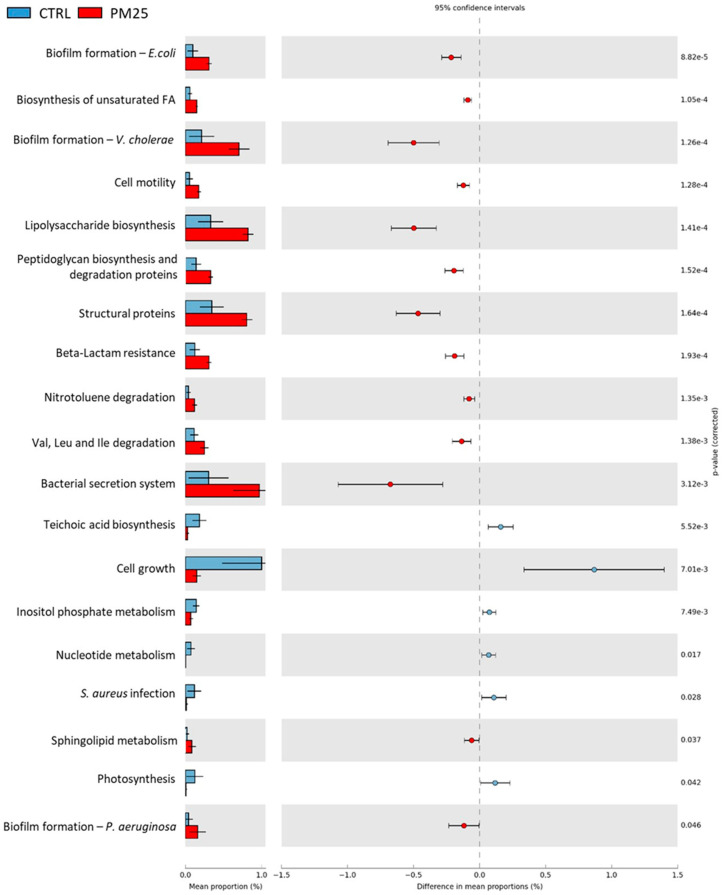
Comparison of the relative abundance of the PICRUSt functional profile of the gut microbiota between the control and PM25 experimental groups. Only the predicted functional pathways that differ significantly (*p* < 0.05) are shown.

**Table 1 microorganisms-12-00744-t001:** Diet formulations.

INGREDIENTS (%)	A	B	C
Fish meal	20.00	10.00	10.00
Poultry by-product	0.00	20.00	25.00
Insect meal (Mutatec)	0.00	5.00	0.00
Exuviae	0.00	0.00	0.50
Rapeseed meal	11.00	11.00	12.50
Soybean meal	11.00	11.00	11.00
Pure starch	13.00	14.00	14.00
Corn gluten meal	21.90	7.90	7.90
Vital wheat gluten	6.00	6.00	6.00
Soy protein concentrate	6.00	4.00	2.00
Soybean oil	3.00	3.00	3.00
Fish oil	3.00	3.00	3.00
DL-methionine	0.80	0.80	0.80
Emulsifier (E484)	0.30	0.30	0.30
Monoammonium phosphate	1.40	1.40	1.40
Lysine HCl	0.80	0.80	0.80
Vitamin and mineral premix ^a^	0.60	0.60	0.60
Antioxidant premix ^b^	0.10	0.10	0.10
Taurine	1.00	1.00	1.00
Hydrolyzed fish protein	0.00	0.00	0.00
Stay C 35%	0.10	0.10	0.10
	100.000	100.00	100.00

^a^ Vitamin and mineral premix (quantities in 1 kg of mix): vitamin A, 1200 mg; vitamin D3, 20 mg; vitamin C, 25,000 mg; vitamin E, 15,000 mg; inositol, 15,000 mg; niacin, 12,000 mg; choline chloride, 6000 mg; calcium pantothenate, 3000 mg; vitamin B1, 2000 mg; vitamin B3, 2000 mg; vitamin B6, 1800 mg; biotin, 100 mg; Mn, 9000 mg; Zn, 8000 mg; Fe, 7000 mg; Cu, 1400 mg; Co, 160 mg; I, 120 mg; anticaking agents and carrier, adding up to 1000 g. ^b^ Antioxidant premix: tocopherol; propyl gallate; butylated hydroxytoluene (BHT); tertiary butylhydroquinone (TBHQ).

**Table 2 microorganisms-12-00744-t002:** Proximate composition of diets.

	A	B	C
Gross energy (MJ/kg)	17.34	17.55	17.58
DE (MJ/kg)	15.60	15.66	15.66
DE (%)	89.98	89.23	89.09
Crude fat (g/100 g)	10.63	10.77	10.82
Crude Protein (g/100 g)	45.08	45.00	45.04
DP (%)	40.17	39.35	39.28
Digestible protein (%)	89.12	87.44	87.22
Fish protein (%)	13.20	6.60	6.60
Animal protein (%)	13.20	22.98	23.85
FP/TP (%)	29.28	14.67	14.65
DP/DE (mg/Kj or g/MJ)	25.75	25.14	25.09
AP/TP (%)	29.28	51.06	52.96
Fiber (g/100 g)	3.03	3.41	3.04
EI (g/100 g)	35.48	32.29	33.01
Amido (g/100 g)	15.95	15.55	15.55
NSP (g/100 g)	22.57	21.15	20.51
Protein-to-lipid ratio	4.50	4.18	4.16
Ash (g/100 g)	6.38	7.53	8.09
DE (kcal/kg)	3729.34	3741.83	3742.61
Crude En (kcal/kg)	4304.66	4243.40	4233.13

Notes: FP/TP: Fish protein/Total protein, DP/DE: digestible protein/digestible energy, AP/TP: animal protein/total protein, NSP: non-starch polysaccharides.

**Table 3 microorganisms-12-00744-t003:** Growth performance of seabass fed with three experimental diets. Different letters in the same row indicate a significant difference between the mean values (*p* < 0.05; N(A) = 70; N(B) = 67; N(C) = 69).

	CTRL	PM20	PM25
IBW (g)	52.88 ± 7.24	52.16 ± 6.14	52.04 ± 5.13
FBW (g)	135.88 ± 20.59 ^a^	120.26 ± 19.76 ^b^	128.37 ± 23.01 ^ab^
SGR (% day^−1^)	1.1	0.96	1.03
FCR	1.76	2.17	1.94

Notes: Initial body weight (IBW); final body weight (FBW); standard growth rate (SGR); feed conversion ratio (FCR). Values are given as the mean ± standard deviation. Different letters in the same row indicate a significant difference between the mean values (*p* < 0.05; N(A) = 70; N(B) = 67; N(C) = 69).

**Table 4 microorganisms-12-00744-t004:** Intestinal morphometric parameters of seabass fed experimental diets.

	CTRL	PM20	PM25	*p*-Value
Proximal intestine				
ViH (μm)	1446.26 ± 61.90	1434.69 ± 44.01	1445.89 ± 54.71	0.985
ViW (μm)	259.24 ± 11.65	244.82 ± 20.69	223.73 ± 22.12	0.4071
LPW (μm)	54.19 ± 3.21	51.74 ± 2.76	51.93 ± 4.38	0.862
MT (μm)	161.22 ± 6.35	169.39 ± 8.14	177.17 ± 8.49	0.29
Distal intestine				
ViH (μm)	1008.78 ± 21.50	1064.90 ± 37.14	1035.87 ± 31.11	0.435
ViW (μm)	78.35 ± 2.57	83.18 ± 3.05	84.79 ± 3.42	0.300
LPW (μm)	28.44 ± 1.00	29.64 ± 1.42	27.3 ± 1.78	0.585
MT (μm)	352.65 ± 14.93	365.94 ± 18.56	332.02 ± 9.22	0.266

Notes: villus height (ViH), villus width (ViW), lamina propria width (LPW), and muscular thickness (MT). Values are expressed as mean ± standard error.

**Table 5 microorganisms-12-00744-t005:** Results of histological scoring of the liver.

	CTRL	PM 20%	PM 25%
HV	2.00	1.67	2.00
ND	2.00	1.33	1.67
NS	1.00	1.00	1.00
CH	1.00	1.00	1.00
Total	6.00	5.00	5.67

Notes: HV, Hepatocytes vacuolization; ND, nuclear displacement; NS, irregular nuclei shapes; CH, cellular hypertrophy.

**Table 6 microorganisms-12-00744-t006:** Volatile SCFA content in fecal samples from three test groups. The results are given in mmol/L, mean ± SD (N = 6).

	Acetate (C 2:0)	Propionate (C 3:0)	Butyrate (C 4:0)
CTRL	14.67 ± 0.66 ^b^	5.93 ± 0.62 ^c^	2.13 ± 0.27 ^b^
PM20	17.05 ± 0.94 ^a^	6.43 ± 0.46 ^b^	2.00 ± 0.13 ^b^
PM25	15.19 ± 0.42 ^b^	7.41 ± 0.18 ^a^	3.03 ± 0.13 ^a^

Notes: The results are given in mmol/L, mean ± SD (N = 6). The lowercase letters (a–c) along the column indicate significant differences at *p* < 0.05.

**Table 7 microorganisms-12-00744-t007:** Alpha diversity analysis of bacterial communities associated with feeds. The lowercase letters (a,b) along the row indicate significant differences at the *p*-value given in the last column.

	A	B	C	*p*-Value
Observed OTUs	1325 ± 216 ^b^	1721 ± 33 ^a^	1224 ± 168 ^b^	0.019
Chao1	1541 ± 250 ^ab^	1893 ± 47 ^a^	1382 ± 226 ^b^	0.047
Faith-PD	21.7 ± 4.9	23.9 ± 1.3	21.8 ± 3.3	>0.05
Shannon	5.8 ± 0.4 ^b^	7.1 ± 0.1 ^a^	5.9 ± 0.2 ^b^	0.004
Simpson	0.92 ± 0.02	0.95 ± 0.00	0.92 ± 0.00	>0.05

**Table 8 microorganisms-12-00744-t008:** Result of the alpha diversity analysis of the microbial communities in the gut.

	CTRL	PM20	PM25	*p*-Value
Observed OTUs	1144 ± 380	756 ± 329	945 ± 445	>0.05
Chao1	1276 ± 440	914 ± 386	1086 ± 471	>0.05
Faith-PD	18.5 ± 6.7	15.8 ± 5.6	18.5 ± 5.9	>0.05
Shannon	6.20 ± 1.06	4.69 ± 0.65	5.18 ± 1.23	0.048
Simpson	0.93 ± 0.04	0.89 ± 0.02	0.89 ± 0.05	>0.05

**Table 9 microorganisms-12-00744-t009:** Validation of the PCoA analysis.

Unweighted UniFrac PCoA					
ANOSIM					
		Sample size	Permutations	R	*p*-value
CTRL	PM20	13	999	0.316	0.005
CTRL	PM25	14	999	0.312	0.005
PM20	PM25	13	999	0.184	0.037
CTRL	A	10	999	0.496	0.030
PM20	B	9	999	0.728	0.014
PM25	C	10	999	0.639	0.007
PERMANOVA					
		Sample size	Permutations	pseudo-F	*p*-value
CTRL	PM20	13	999	1.922	0.010
CTRL	PM25	14	999	1.979	0.003
PM20	PM25	13	999	1.567	0.024
CTRL	A	10	999	2.486	0.013
PM20	B	9	999	4.473	0.012
PM25	C	10	999	3.749	0.007
Weighted UniFrac PCoA					
ANOSIM					
		Sample size	Permutations	R	*p*-value
CTRL	PM20	13	999	0.947	0.003
CTRL	PM25	14	999	0.892	0.002
PM20	PM25	13	999	−0.009	0.437
CTRL	A	10	999	1	0.009
PM20	B	9	999	1	0.013
PM25	C	10	999	1	0.009
PERMANOVA					
		Sample size	Permutations	pseudo-F	*p*-value
CTRL	PM20	13	999	38.427	0.002
CTRL	PM25	14	999	26.332	0.001
PM20	PM25	13	999	1.441	0.233
CTRL	A	10	999	23.772	0.012
PM20	B	9	999	268.127	0.012
PM25	C	10	999	75.875	0.011

**Table 10 microorganisms-12-00744-t010:** The list of genera of intestinal bacteria that differ between CTRL and PM20 fish.

Phylum	Class	Order	Family	Genus	CTRL (%)	SD (%)	PM20 (%)	SD (%)	*p*-Values
Actinobacteriota	Actinobacteria	Corynebacteriales	Mycobacteriaceae	Mycobacterium	0.69	0.51	0.04	0.05	0.0210
Bacteroidota	Bacteroidia	Flavobacteriales	Flavobacteriaceae	Ulvibacter	0.11	0.08	0.01	0.02	0.0196
Firmicutes	Bacilli	Staphylococcales	Staphylococcaceae	Staphylococcus	36.87	24.78	1.14	1.04	0.0123
Firmicutes	Bacilli	Bacillales	Bacillaceae	Geobacillus	0.84	0.62	0.00	0.00	0.0168
Firmicutes	Clostridia	Clostridiales	Clostridiaceae	Clostridium_sensu_stricto_7	1.23	0.96	0.01	0.01	0.0208
Firmicutes	Clostridia	Clostridiales	Clostridiaceae	Clostridium_sensu_stricto_1	0.53	0.36	0.07	0.14	0.0228
Firmicutes	Clostridia	Peptostreptococcales-Tissierellales	Peptostreptococcales-Tissierellales	Anaerosalibacter	1.08	0.89	0.02	0.03	0.0259
Firmicutes	Clostridia	Clostridiales	Clostridiaceae	Clostridium_sensu_stricto_18	0.28	0.24	0.01	0.01	0.0364
Firmicutes	Bacilli	Bacillales	Bacillaceae	Oceanobacillus	0.25	0.21	0.04	0.04	0.0494
Firmicutes	Bacilli	Lactobacillales	Lactobacillaceae	Lactobacillus	22.47	22.09	0.35	0.36	0.0496
Fusobacteriota	Fusobacteriia	Fusobacteriales	Fusobacteriaceae	Cetobacterium	1.47	0.81	0.00	0.01	0.0044
Proteobacteria	Gammaproteobacteria	Vibrionales	Vibrionaceae	Photobacterium	6.66	5.30	43.79	10.80	0.0002
Proteobacteria	Gammaproteobacteria	Vibrionales	Vibrionaceae	Vibrio	6.70	11.52	44.16	13.29	0.0006

**Table 11 microorganisms-12-00744-t011:** The list of genera of intestinal bacteria that differ between CTRL and PM25 fish.

Phylum	Class	Order	Family	Genus	CTRL (%)	SD (%)	PM25 (%)	SD (%)	*p*-Value
Firmicutes	Clostridia	Clostridiales	Clostridiaceae	Clostridium sensu stricto 1	0.53	0.36	0.06	0.08	0.0188
Firmicutes	Bacilli	Paenibacillales	Paenibacillaceae	Paenibacillus	9.33	7.21	0.06	0.07	0.0197
Firmicutes	Bacilli	Bacillales	Bacillaceae	Geobacillus	0.84	0.62	0.04	0.05	0.0209
Firmicutes	Bacilli	Staphylococcales	Staphylococcaceae	Staphylococcus	36.87	24.78	6.14	8.80	0.0226
Firmicutes	Bacilli	Bacillales	Bacillaceae	Oceanobacillus	0.25	0.21	0.00	0.01	0.0288
Firmicutes	Clostridia	Peptostreptococcales-Tissierellales	Peptostreptococcales-Tissierellales	Anaerosalibacter	1.08	0.89	0.06	0.09	0.0300
Firmicutes	Bacilli	Mycoplasmatales	Mycoplasmataceae	Mycoplasma	0.02	0.05	1.23	1.14	0.0413
Firmicutes	Clostridia	Clostridiales	Clostridiaceae	Clostridium sensu stricto 18	0.28	0.24	0.03	0.05	0.0452
Fusobacteriota	Fusobacteriia	Fusobacteriales	Fusobacteriaceae	Cetobacterium	1.47	0.81	0.08	0.16	0.0052
Proteobacteria	Gammaproteobacteria	Vibrionales	Vibrionaceae	Vibrio	6.70	11.52	43.21	30.77	0.0273
Proteobacteria	Gammaproteobacteria	Vibrionales	Vibrionaceae	Photobacterium	6.66	5.30	34.77	26.22	0.0393

## Data Availability

All raw sequencing data were submitted to the European Nucleotide Archive (EBI ENA) public database under accession code PRJEB70800.

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
