# Peer review of "The Replacement of Fish Meal with Poultry By-Product Meal and Insect Exuviae: Effects on Growth Performance, Gut Health and Microbiota of the European Seabass, Dicentrarchus labrax"

_microorganisms, 2024, doi:10.3390/microorganisms12040744_

Round 1

Reviewer 1 Report

Comments and Suggestions for Authors

The paper entitled "The replacement of fish meal with poultry by-product meal and insect exuviae: effects on growth performance, gut health and microbiota of the European seabass, Dicentrarchus labrax" (Microorganisms 2924489) explored the effect of poultry by-product meal (PBM) and its combination with insect exuviae meal as a fishmeal alternative on growth, visceral morphology, gut microbiota, and short chain fatty acid production in European seabass. Dietary PBM and the combination of PBM and insect exuviae meal showed no negative influence on the most parameters detected in European seabass when the replacement of fishmeal reached up to 50%. Moreover, the diet containing 25% PBM exhibited superior growth data. Results from this study highlight the feasibility of poultry by-product meal alone or in combination with insect exuviae meal to improve growth and health of European seabass and other fish species and provide the technical support for developing new, effective, and sustainable animal-derived protein source to fishmeal in aquafeed.

Findings in this manuscript are valuable and meaningful.

In this case, the paper should be streamlined, particularly the content of introduction and discussion parts, and be proofread throughout for improving readability.

Major comments:

1.In the section of "1.Introduction", the description on poultry meal, insect meal, and their applications in aquafeed is too detailed and distracting. For example, in Line 54-56, it contains several sentences that could be deleted without influencing the meaning of the whole paragraph. It is suggested that the main text in Line 52-75 is simplified and consolidated into one paragraph. The statement in Line 84-109 contains the same error. Please streamline and compress the presentation of known and/or expected correlative findings for a clear focus on background in this study.

2. Regarding the Table 1 (Line 128-130), no information is provided as the source or quality of "Minerals and Vitamins" and "Premix antioxidants" in the experimental feed? The authors should give the relevant information in the legend of Table 1.

3. Prior to the feeding experiments, aquatic animals are cultured for one week to adapt to the laboratory conditions. What feed were used in European seabass during the acclimating period? Control diet or other feed? The authors should provide the relevant information to "2.3. Feeding trial and sampling" section. 

4. The main text of discussion part contains the repeating and long statements. Please streamline and written the section of "Discussion" with emphasis for better clarifying your findings in this study.

Minor comments:

1. Check the symbols or codes for volume unit in this study according to the information to related guides. For example, Line 144, please replace "g.L-1" with "g/L". Also, there were similar errors in the other part of this manuscript.

2. The legends of some tables should be indicated below tables, such as Table 1, Table 6. Also, the tables (Table 8-10) that display much information should be provided as "Supporting Information". Please check the whole text and revise accordingly.

3. In "3.Results" section, there was some redundancy in the descriptions of results. The contents of this section should be simplified. For example, it would be preferable to directly delete the statement in Line 246-247.

Other errors were presented in the PDF file.

Therefore, this manuscript will be accepted after minor revision.

Comments on the Quality of English Language

This manuscript (Microorganisms ID 2924489) entitled "The replacement of fish meal with poultry by-product meal and insect exuviae: effects on growth performance, gut health and microbiota of the European seabass, Dicentrarchus labrax" is valuable for aquaculture and aquafeed. But the description of the whole paper is lengthy and sometimes a little bit redundant, particularly the section of "Introduction" and "Discussion". Also there are still several mistakes, such as the presenting symbols for volume unit.

Reviewer 2 Report

Comments and Suggestions for Authors

This is a well-structured and documented article that demonstrates that by combining poultry by-product meal (PBM) diets with insect or exuviae meal diets, we can replace a significant portion of fish meal (FM) without negatively impacting (or compromising) sea bass growth, gut health, or welfare.

One of the study's primary advantages is that a range of analyses were used to validate the effects of FM replacement.

The study replicated previous findings on the favourable effects of combined PBM and insect or exuviae diets and showed that organic wastes may be used effectively to boost sea bass farming productivity.

Given that the article is well-written and easily reading, I propose publishing this in its present form.

Minor changes

L143-144. Change gL-1 to gL-1

Author Response

Thank you for your review. We have made the necessary correction as per your suggestion.